# Sparse Representation and SVM Diagnosis Method for Inter-Turn Short-Circuit Fault in PMSM

**Siyuan Liang** [1,2], **Yong Chen** [1,2,*] , **Hong Liang** [1,2,3] **and Xu Li** [4]

1 School of Automation Engineering, University of Electronic Science and Technology of China, Chengdu 611731, China; 13308010916@163.com (S.L.); liangh12@sina.com (H.L.)
2 Institute of Electric Vehicle Driving System and Safety Technology, University of Electronic Science and Technology of China, Chengdu 611731, China
3 Unit 69031 of the People's Liberation Army of China, Urumqi 830000, China
4 Chongqing Changan Automobile Co Ltd., Chongqing 400023, China; lixu@changan.com.cn
* Correspondence: ychencd@uestc.edu.cn

**Abstract:** Permanent magnet synchronous motors (PMSM) has the advantages of simple structure, small size, high efficiency, and high power factor, and a key dynamic source and is widely used in industry, equipment and electric vehicle. Aiming at its inter-turn short-circuit fault, this paper proposes a fault diagnosis method based on sparse representation and support vector machine (SVM). Firstly, the sparse representation is used to extract the first and second largest sparse coefficients of both current signal and vibration signals, and then they are composed into four-dimensional feature vectors. Secondly, the feature vectors are input into the support vector machine for fault diagnosis, which is suitable for small sample. Experiments on a permanent magnet synchronous motor with artificially set inter-turn short-circuit fault and a normal one showed that the method is feasible and accurate.

**Keywords:** fault diagnosis; inter-turn short circuit; sparse representation; support vector machine; PMSM

## 1. Introduction

Permanent magnet synchronous motors (PMSM) has the advantages of simple structure, small size, high efficiency, and high power factor [1], and a key dynamic source and is widely used in industry, equipment and electric vehicle. Once some faults occur in the motor, it can cause serious damage of equipment and property. Therefore, it is important to research the diagnose methods of the faults in PMSM. Motor faults can be broadly classified into mechanical faults, magnetic faults and electrical faults. The inter-turn short-circuit is the most common electrical fault of the motor, whose major source is the insulation problem of the stator windings [2].

Signal processing is one of the commonly used fault diagnosis methods for motor, such as motor current signal analysis (MCSA) [3]. Signal processing methods typically include time domain methods, frequency domain methods, and time-frequency analysis methods. The classical method of frequency domain analysis is fast Fourier transform (FFT) [4]. It can clearly show the frequency distribution of the signal, where the harmonic component can be used as a feature of the fault. In order to overcome the shortcoming of FFT which loss time domain information, time-frequency analysis methods are proposed, such as short-time Fourier transform (STFT) [5], wavelet transform (WT) [6] etc. But the studies before often only process the single signal of current or vibration.

With the development of machine learning and artificial intelligence, lots of data-driven intelligent diagnostic methods have been proposed, such as fuzzy logic (FL), neural networks (NN), and support vector machines (SVM) [7]. They can classify and identify the input data to determine the fault

state of the motor. SVM is an algorithm suitable for small sample which was proposed in 1990s. S. Das et al. combined Park's transformation and continuous wavelet transform (CWT) to extract features, and then import them into SVM for classification [8]. Chen F extracted features by using empirical mode decomposition (EMD), and then optimized the wavelet support vector machine to diagnose [9]. H. Hassani et al. used fuzzy logic systems to combine different SVMs for bearing fault detection [10]. Today, the application of SVM for PMSM fault diagnosis can still be explored.

Sparse representation is a new theory which is the most representative methodology of the linear representation methods [11], so it has been used in fields of image processing, signal processing, computer vision, and pattern recognition [12]. In signal processing, it can compress signals and extract the essential characteristics of signals, so it has also been used for mechanical fault diagnosis in recent years. Sparse representation has been used in feature extraction, signal denoising, and fault classification in the field of fault diagnosis. N. Chai et al. used resonance-based sparse signal decomposition (RSSD) to remove the interference from the gear meshing-related components in the current signal [13]. L. Ren et al. used a joint model based on sparse representation classification (SRC) [14] and SVM in the roller bearing fault diagnosis [15]. Jing Hou et al. studied the feature extraction method of power electronic circuits based on sparse representation, which used matching pursuit (MP) algorithm to obtain the maximum sparse coefficient as the feature [16]. Instead of the constructed dictionary, B. Yang used a dictionary learning algorithm to improve the application of sparse representation in diagnosis of wind turbine generator bearing fault [17]. However, most studies used sparse representation to process vibration signals to diagnose the mechanical faults. There are few studies specifically researching the electrical faults of PMSM.

Aiming at the problem of inter-turn short circuit fault in PMSM, this paper proposes a fault diagnosis method based on sparse representation and SVM. The sparse representation is used to extract the first and second largest sparse coefficients of both current signal and vibration signals, and they are composed into four-dimensional feature vectors, which are input into the SVM for training. At last, the trained SVM model is used for fault diagnosis. Section 2 introduces the principle and solution algorithm of sparse representation used in this paper, as well as the features selected in this work. Section 3 introduces the principle of SVM. In Section 4, two PMSM were used in the experiment: one motor was manually set the inter-turn short-circuit fault and the other was normal. The experiment results showed that the method is feasible.

## 2. Sparse Representation

The principle of sparse representation theory is to represent the original signal as a sparse linear combination of the dictionary atoms in an overcomplete dictionary matrix [11]. Therefore, it can extract the basic information from the signal, and avoid the interference of small noise in the signal, such as additional vibration caused by rotor imbalance or distortion of the measurement process. The main idea of sparse representation theory is to concentrate the energy of feature information into a few elements [17]. Therefore, in this paper, sparse representation is used to extract fault features. For an input signal $x = [x_1, x_1, \cdots, x_n]^T$ of length $n$:

$$x = Da \tag{1}$$

$$x_i = \sum_{i=0}^{m} \alpha_i d_i \tag{2}$$

where $D = [d_1, d_2, \cdots, d_m]$ is the dictionary matrix, the column vector $d_i$ is a dictionary atom and $||d_i = 1||$. The dictionary is an overcomplete dictionary when $m > n$. $\alpha = [\alpha_1, \alpha_2, \cdots, \alpha_m]^T$ is the solution to the sparse representation of the original signal, and $\alpha_i$ is the sparse representation coefficient. Use the $l_0$ norm to indicate the sparse degree of the sparse coefficients:

$$||\alpha||_0 = \lim_{p \to 0} \sum_{i=1}^{n} |\alpha_i|^p \tag{3}$$

Obviously, the $l_0$ norm is actually the number of non-zero elements in $\alpha$.

The commonly used optimization algorithm based on $l_0$ norm is matching pursuit (MP) [18]. Firstly, it chooses an atom that matches the input signal best from the dictionary matrix and calculates the residual. Then it continues to select the atom that matches the residual signal best, and iterates over and over like this until it meets the requirement. However, in MP, the residual signal is only orthogonal to the dictionary atom selected in this iteration, and does not necessarily have orthogonality with all dictionary atoms, which will increase the number of iterations, and the result may not be the optimal solution. In order to solve this problem, the orthogonal matching pursuit (OMP) algorithm was proposed. The workflow of OMP is shown in Table 1:

**Table 1.** Workflow of orthogonal matching pursuit (OMP).

| **Input: signal $x$, dictionary matrix $D$, sparse degree K** |
| --- |
| (1) Initialize residual signal $R_0 = x$, index set $\Lambda_0 = \varnothing$, matrix $A_0 = \varnothing$, number of iterations $t = 1$. |
| (2) Select the atom $d_t$ that best matches the residual signal in the dictionary. The selection condition is $|\langle R_{t-1}, d_t \rangle| = \sup|\langle R_{t-1}, d_k \rangle|$ and the column number of the selected atom satisfies that $\lambda_t \notin \Lambda_{t-1}$. |
| (3) Update the index set $\Lambda_t = \Lambda_{t-1} \cup \{\lambda_t\}$ with $\lambda_t$, and update the matrix $A_t = A_{t-1} \cup d_t$. |
| (4) Calculate the least squares solution $\hat{\alpha} = argmin||x - A_t\alpha_t|| = (A_t^T A_t)^{-1} A_t^T x$. |
| (5) Update residual $R_t = x - A_t\hat{\alpha} = x - A_t(A_t^T A_t)^{-1} A_t^T x$. |
| (6) Update $t = t + 1$ and repeat steps 2–5 until $t = K$. |
| Output: sparse representation coefficients $\alpha$ |

As for the choice of dictionary matrix, discrete cosine transform (DCT) is suitable for stationary signals and has higher computational efficiency than Fourier transform, so it is used to construct a dictionary matrix. The effective information of the result by DCT is mostly concentrated in the low frequency, which can remove the redundant information of the signal while retaining the basic information.

In this paper, the sparse coefficients of the current signal and the vibration signal are obtained by sparse representation respectively, and the first and second largest value in the respective sparse coefficients $\alpha$ are selected to be constructed into four-dimensional feature vectors, which will be input into the SVM.

## 3. SVM

SVM is a two-class model whose purpose is to find a hyperplane to segment the sample space. The principle of segmentation is to enable the hyperplane to classify the samples correctly and maximize the margin between the samples and the hyperplane. The samples closest to the hyperplane that determines the position of the hyperplane are support vectors.

Let the sample points for training be $(x_i, y_i)$, $i = 1, 2, \cdots, l$. The input is $x_i \in R^d$, and the output is $y_i \in \{-1, 1\}$. If let $w$ be the normal vector of the hyperplane and $b$ be the displacement, the equation of the hyperplane is:

$$w^T x + b = 0 \tag{4}$$

The margin is equal to the projection of the difference between two different types of support vectors on $w$, which is $\frac{2}{||w||}$. Maximizing the classification margin is equal to the following:

$$\min_{w,b} \frac{||w||^2}{2}, s.t. y_i \left( w^T x_i + b \right) \geq 1 \tag{5}$$

Its Lagrange function is:

$$L(w, b, \alpha) = \frac{\|w\|^2}{2} + \sum_{i=1}^{m} \alpha_i \left(1 - y\left(w^T x_i + b\right)\right) \tag{6}$$

where $\alpha_i \geq 0$ is the Lagrange multiplier. Let the partial derivatives of $L(w, b, \alpha)$ with respect to $w$ and $b$ respectively equal to 0, and then the following equations can be obtained:

$$\begin{cases} w = \sum\limits_{i=1}^{m} \alpha_i y_i x_i \\ \sum\limits_{i=1}^{m} \alpha_i y_i = 0 \end{cases} \tag{7}$$

Substitute the above equation into the original function, and the original question becomes:

$$\max_{\alpha} \sum_{i=1}^{m} \alpha_i - \frac{1}{2} \sum_{i=1}^{m} \sum_{i=1}^{m} \alpha_i \alpha_j y_i y_j x_i x_j, s.t. \sum_{i=1}^{m} \alpha_i y_i = 0, \alpha_i \geq 0, i = 1, 2, \cdots, m \tag{8}$$

From this, the decision function corresponding to the segmentation hyperplane equation can be solved, which is:

$$f(x) = sign\left(w^T x + b\right) = sign(\sum_{i=1}^{m} \alpha_i y_i x_i^T x + b) \tag{9}$$

In this paper, the input vector of sample point is four-dimensional, which means $x_i \in R^4$. As for the output $y_i \in \{-1, 1\}$, 1 represents the signal of the normal motor and -1 represents the signal of the inter-turn short circuit motor. First, the training sets where fault types are known are input into the SVM to solve the decision function. Then, input the test set to test the diagnosis effect of the inter-turn short-circuit fault in PMSM based on SVM.

## 4. Experimental Results and Analysis

### 4.1. Experiment Setup

The experimental platform for fault diagnosis of PMSM is shown in Figure 1. The current sensor and the vibration sensor collect signals from the motors, and data acquisition (DAQ) card uploads the signal data to the computer for the subsequent research on the algorithm.

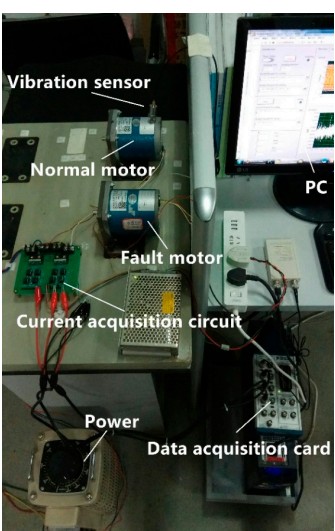

**Figure 1.** The experimental platform for fault diagnosis of permanent magnet synchronous motors (PMSM).

The experimental platform includes two low speed PMSMs (90TDY115-2B), current sensors (The experimental platform), piezoelectric accelerometer (CA-YD-186) and DAQ card (NI-PCI6225). One of them is a normal motor and the other one is artificially set to a 15% stator inter-turn fault. These current sensors acquire the stator current of the motors and transform the current signal into voltage signal that is convenience for the host computer to deal with. Piezoelectric accelerometer gets the vibration information and its output is amplified by signal conditioner (YE3832). In the host computer, A DAQ card, NI-PCI6225, was installed, to which a junction box, BNC-2110, was connected. They can upload the data from sensors to the host computer. After the data were saved, the algorithm in this paper was implemented in MATLAB (2015b, MathWorks, Natick, MA, USA., 1984).

### 4.2. Feature Extraction Based on Sparse Representation

In this work, 40 sets of single-phase current signals were collected from the motors, of which 20 sets were from normal motor and 20 sets were from the fault one. According to Nyquist's sampling law, the sampling frequency $f_s$ was set to 1024 Hz, and the number of sampling points $N$ was 4096, which means the time of each set was 4 s. The same applied to the vibration signal. Among them, one of the collected normal currents and one of fault currents are shown in Figure 2.

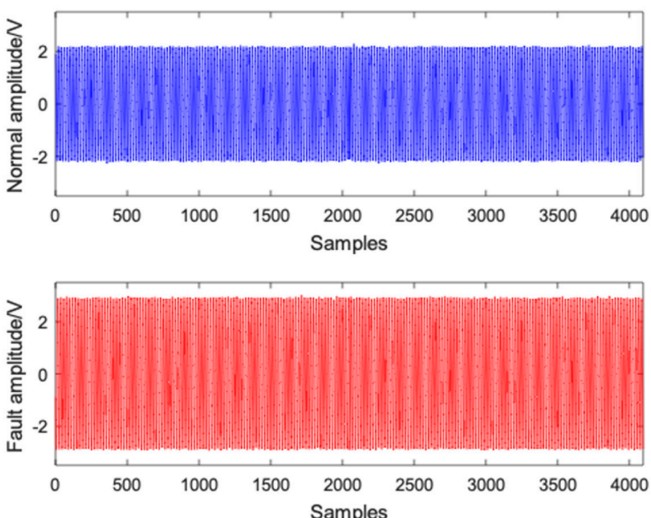

**Figure 2.** Two of the collected current signals.

Their spectrum diagrams obtained by FFT are shown in Figure 3.

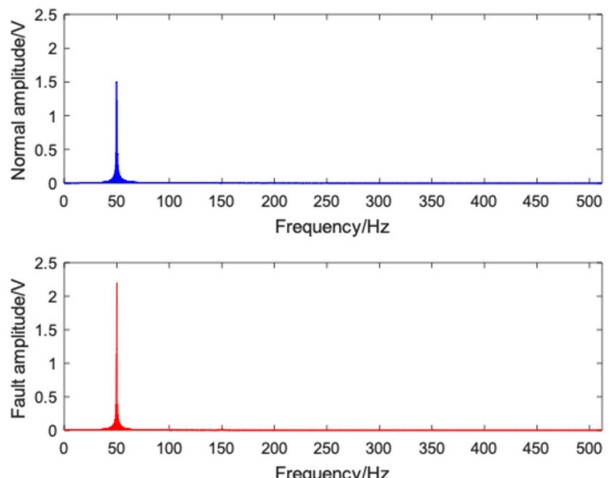

**Figure 3.** Spectrum diagrams of the collected current signals.

Besides, one of the collected normal vibration signals and one of fault vibration signals are shown in Figure 4.

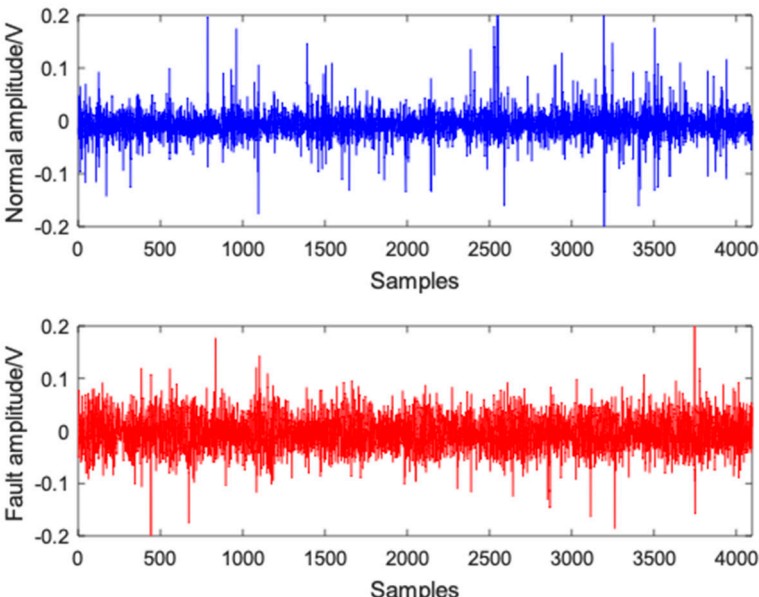

**Figure 4.** Two of the collected vibration signals.

Their spectrum diagrams obtained by FFT are shown in Figure 5.

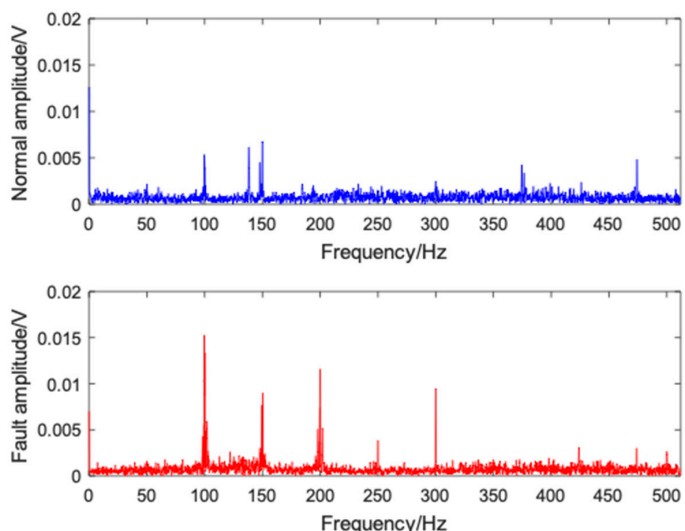

**Figure 5.** Spectrum diagrams of the collected vibration signals.

From Figures 2–4, It can be visually seen from the waveform diagram of the time domain that the amplitude of the fault signals is higher than that of the normal signals. In the spectrum diagrams, the amplitude of the fundamental frequency $f_s = 50$ Hz of the fault current signal is also higher than the amplitude of the normal In addition, there is a tiny harmonic component at $3f_s$ in the fault current signal, and in the vibration signal, some harmonic components at $4f_s$, $6f_s$ and so on can be found. It is in agreement with the previous study [19], which confirms the validity of the acquired signals.

Next, the OMP algorithm is used to obtain their sparse coefficients by DCT overcomplete dictionary. According to the length of the signal, the size of the overcomplete dictionary matrix is set to $n \times m = 4096 \times 16384$. The iterative process of the above normal current signal in OMP is shown in Figure 6.

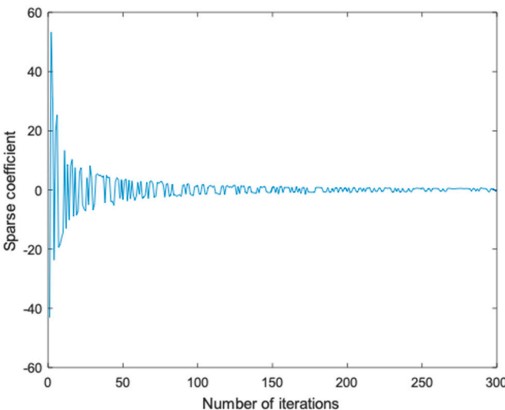

**Figure 6.** Iterative process of one normal current signal.

It can be seen from the figure that the sparse coefficients obtained after the number of iterations is more than 100 times are too small, and should be regarded as noise that should be ignored. Therefore, let the target sparse degree $K = 100$. Next, take the above four sets of signals as examples. After the OMP, the sparse coefficients $\alpha = [\alpha_1, \alpha_2, \cdots, \alpha_{16384}]^T$ obtained by the sparse representation of the two sets current signals are shown in Figure 7.

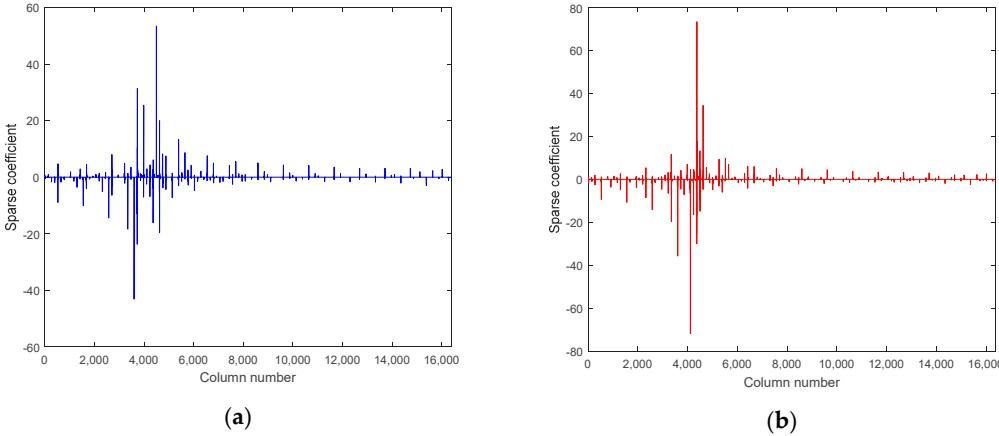

**Figure 7.** Sparse coefficients of two current signals (**a**) Normal current; (**b**) Fault current.

Besides, the sparse coefficients obtained by the sparse representation of the two sets of vibration signals are shown in Figure 8.

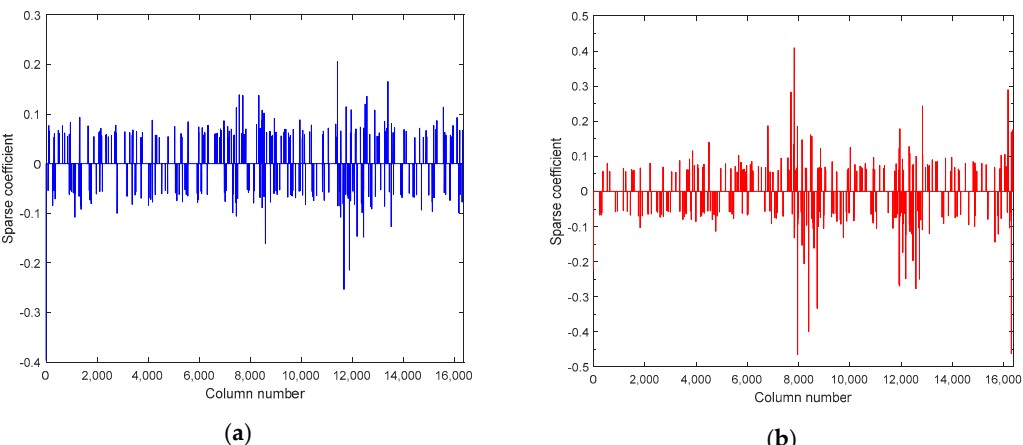

**Figure 8.** Sparse coefficients of two vibration signals (**a**) Normal current; (**b**) Fault current.

As shown in the figure, the values of most small sparse coefficients of the fault signal and the normal signal is almost the same, but the difference between the large sparse coefficients is obvious. The amplitude of large sparse coefficients of fault signal are usually higher than them of the normal signal, especially the maximum sparse coefficient. Therefore, the maximum sparse coefficient can be taken as a feature. However, only using one feature which is the maximum sparse coefficient is inaccurate. For example, as for the sparse coefficients of another normal current signal which is shown in Figure 9, its maximum sparse coefficient is relatively large, which will confuse the diagnosis result. But even in this case, the second largest sparse coefficient of the signal is still smaller than the second largest sparse coefficient of the fault signal. For the sake of accuracy, the maximum sparse coefficient and the second largest sparse coefficient should be taken together as features.

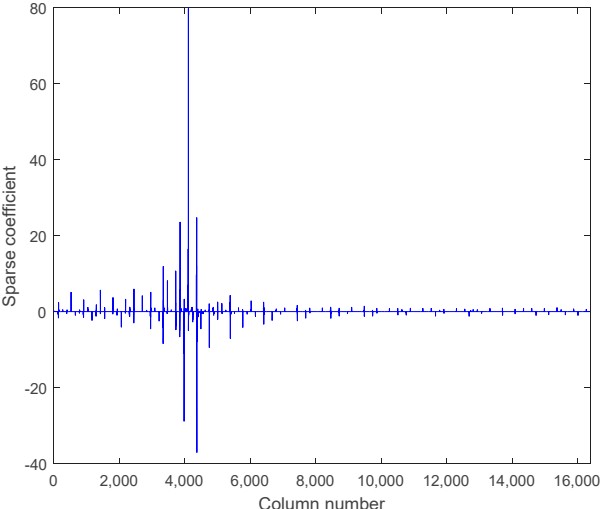

**Figure 9.** Sparse coefficients of a set of normal current signal.

After the above-mentioned OMP algorithm is used to obtain the sparse coefficients of 40 sets current signals and the sparse coefficients of 40 sets vibration signals, select the largest and second largest sparse coefficients in them as features. The first and second features extracted from the 20 sets of normal current signals and 20 sets of fault current signals are shown in Figure 10.

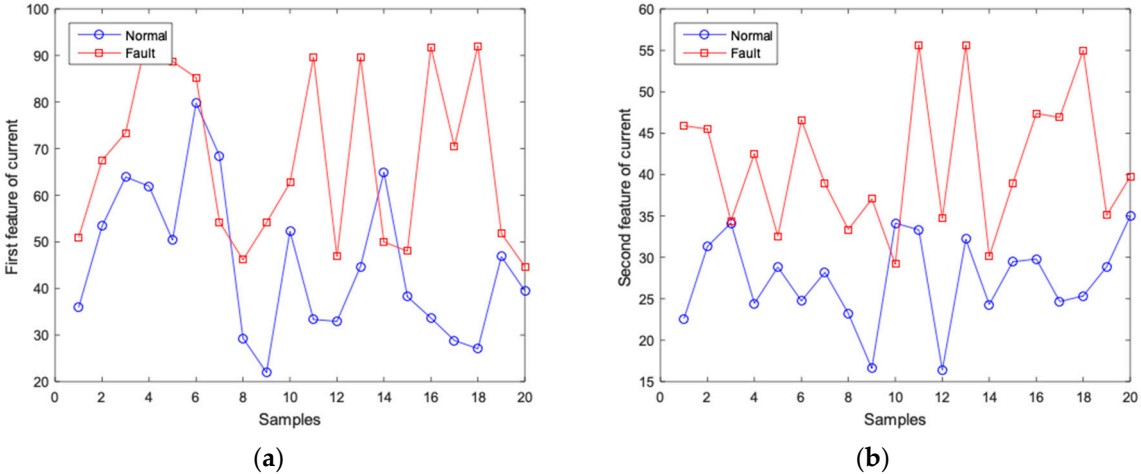

(**a**)                    (**b**)

**Figure 10.** Features extracted from the current signals (**a**) First feature of current; (**b**) Second feature of current.

The first and second features extracted from the 20 sets of normal vibration signals and 20 sets of fault vibration signals are shown in Figure 11.

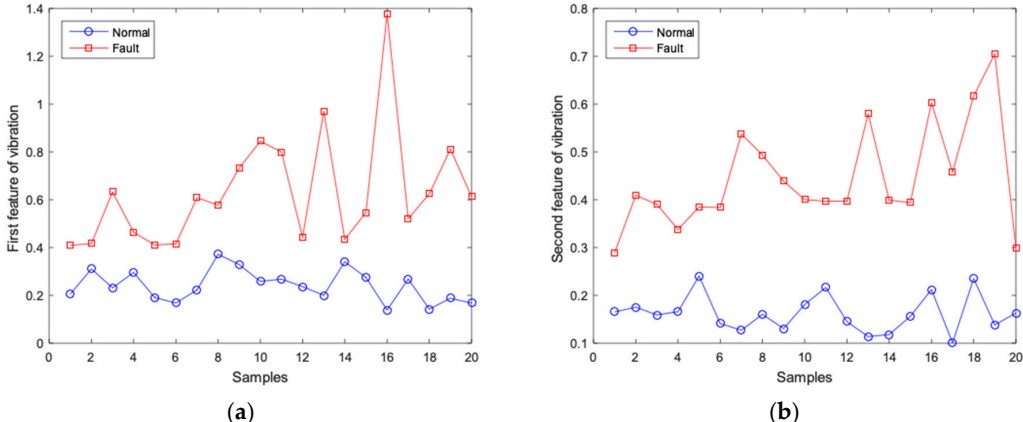

(**a**)  (**b**)

**Figure 11.** Features extracted from the vibration signals (**a**) First feature of vibration; (**b**) Second feature of vibration.

As can be seen from the above figures, the feature values of signals of different fault types are also different. All the first features of fault vibration are higher than 0.4, while the normal features are lower. Besides, the second features of fault vibration are higher than 0.25, while the normal features are lower. Similar phenomena can also be found in the current signals, although not so obvious. The boundary value between the first features of the normal and fault current is about 50, and the boundary value of the second features is about 33. Therefore, combining the different features into one feature vector is suggested so that no information source will be ignored. Next, the SVM will be trained and tested with the feature vectors.

### 4.3. Training and Testing Results of SVM

Mark the extracted features as $F1$, $F2$, $F3$, $F4$, where $F1$ means the first feature of current, $F2$ means the second feature of current, $F3$ means the first feature of vibration, and $F4$ means the second feature of vibration. Then the input become $x_i = [F1_i F2_i F3_i F4_i]$. The 40 feature vectors obtained above are divided into two sets, where 34 sets are training sets and 6 sets are test sets. The number of normal and fault feature vectors in each set is the same.

So, the sample points for training SVM are $(x_i, y_i)$, and $y_i$ is their corresponding fault type, $i = 1, 2, \cdots, 34$. Input them into the SVM to get the support vectors and hyperplane. The sample points and support vectors (SV) are shown in Figure 12 where the two features of current act as the coordinate axes.

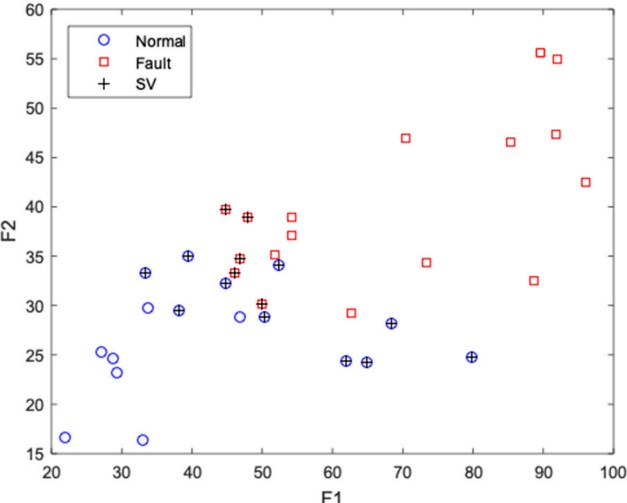

**Figure 12.** Sample points and support vectors by $F1$ and $F2$.

The sample points and support vectors (SV) are also shown in Figure 13 where the two features of vibration act as the coordinate axes.

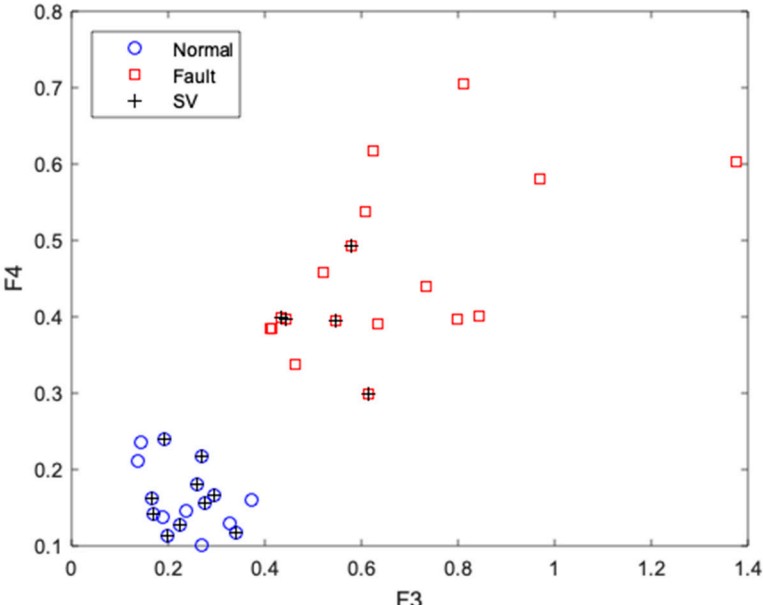

**Figure 13.** Sample points and support vectors by *F*3 and *F*4.

As can be seen, in the two figures, the sample points of the normal signals are concentrated in the lower left corner, and the sample points of the fault signal are scattered in the upper right corner. After the training is completed, input the 6 test sets to test. The results are shown in Table 2.

**Table 2.** Test results of support vector machine (SVM).

| Sample | *F*1 | *F*2 | *F*3 | *F*4 | Result |
|--------|------|------|------|------|--------|
| 1 | 35.9105 | 22.6089 | 0.2055 | 0.1654 | 1 |
| 2 | 50.9903 | 45.8897 | 0.4094 | 0.2895 | −1 |
| 3 | 53.3843 | 31.3187 | 0.3106 | 0.1755 | 1 |
| 4 | 67.5021 | 45.5020 | 0.4165 | 0.4091 | −1 |
| 5 | 52.2853 | 34.1310 | 0.2317 | 0.1591 | 1 |
| 6 | 73.4088 | 34.3846 | 0.6328 | 0.3905 | −1 |

The accuracy of the above test results is 100%, which proves the features extracted by sparse representations can be used for fault diagnosis of PMSM, and the method proposed in this paper is accurate.

## 5. Discussion

In this paper, a diagnosis method of inter-turn short-circuit fault in PMSM where sparse representation is used to extract features and SVM is used for classification is studied. Previous studies on sparse representation mostly used vibration signals to diagnose mechanical faults of induction motors or gearboxes. In this work, current signals and vibration signals are combined to diagnose electrical faults in PMSM by sparse representation and SVM. Experiments indicate that the chosen features and the proposed method are feasible.

However, among the features, as can be seen from the experimental results, compared with the features of vibration signal, the features of the current are not so obvious, and it is not powerful enough to distinguish between normal and fault signals just by them, which may be due to the selection of the dictionary or the optimization algorithm. Therefore, the result of current signal processed by sparse

representation algorithm needs to be improved. In the future, dictionary learning algorithms should also be studied, in order to be more suitable for the feature extraction of motor signals. In addition, more complicated conditions and more PMSM faults need to be considered.

## 6. Conclusions

Aiming at the problem of inter-turn short circuit fault in PMSM, this paper proposes a fault diagnosis method based on sparse representation and SVM. The sparse representation is used to extract the first and second largest sparse coefficients of both current signal and vibration signals by OMP, and they are composed into four-dimensional feature vectors, which are input into the SVM for training and testing then. Two PMSMs were used in the experiment, one manually set the inter-turn short circuit and one normal. The experiment results showed that the method is feasible and accurate. More advanced sparse representation algorithms and more motor faults need to be researched in the future.

**Author Contributions:** Conceptualization, Y.C.; methodology, Y.C. and S.L.; software, S.L.; validation, S.L. and H.L.; formal analysis, S.L. and Y.C.; investigation, S.L.; resources, Y.C. and S.L.; data curation, H.L. and S.L.; writing-original draft preparation, S.L.; writing—review and editing, S.L. and Y.C.; visualization, S.L.; supervision, Y.C.; project administration, Y.C.; funding acquisition, Y.C.

**Funding:** This research was funded by the National Natural Science Foundation (51607027), the National key R & D Plan Program (2018YFB0106100), the Sichuan Science and Technology support Program (2016GZ0395, 2017GZ0395, and 2017GZ0394), and the Central University basic Research Business funds (ZYGX2016J140 and ZYGX2016J146).

**Conflicts of Interest:** The authors declare no conflict of interest.

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
