# Peer review of "Sparse Representation and SVM Diagnosis Method for Inter-Turn Short-Circuit Fault in PMSM"

_applsci, doi:10.3390/app9020224_

Round 1

Reviewer 1 Report

The paper is well written and clear with a few minor mistakes. My main concern is regarding the contribution. The manuscript looks at the moment as it belongs in a fair conference proceeding rather than a journal. A methodology with step-by-step procedures has been proposed for one particular motor fault. The results obviously (?) are good and that was clear already from the abstract. On the other hand, the work does not give any particular new insight for the problem. What do the different coefficients represent? Will they be robust to varying operating conditions, e.g. change in speed or torque? What more information is found from vibration than current? As vibration seems better, does current in reality add more information (improve classification) compared to vibration alone? There are numerous questions to be answered, while the discussion section (Section 4) is set aside a handful of lines. Moreover, the discussion is not really discussing the results but more a form of summary. There is also a comment/conclusion that the current signal analysis needs to be improved with a given method. This is not really justified as from my point of view the method currently under analysis is not really well understood. Not the method per say, but the results it offers.

In Table 1. Point 5) is A_t missing in the last part of the line before the expression \hat \alpha is inserted from point 4)?

Page 9 (line 194) it is mentioned that normal signals are higher and faulty signals are lower but from Figs. 9 the faulty features appears larger (stronger) than the normal features.

F_i, i={1,..,4} represents the extracted features. Which is what is somewhat vaguely mentioned in the text, but can it be written explicitly at the start of Section 4.3?

Author Response

1. The paper is well written and clear with a few minor mistakes. My main concern is regarding the contribution. The manuscript looks at the moment as it belongs in a fair conference proceeding rather than a journal. A methodology with step-by-step procedures has been proposed for one particular motor fault. The results obviously (?) are good and that was clear already from the abstract. On the other hand, the work does not give any particular new insight for the problem. What do the different coefficients represent? Will they be robust to varying operating conditions, e.g. change in speed or torque? What more information is found from vibration than current? As vibration seems better, does current in reality add more information (improve classification) compared to vibration alone? There are numerous questions to be answered, while the discussion section (Section 4) is set aside a handful of lines. Moreover, the discussion is not really discussing the results but more a form of summary. There is also a comment/conclusion that the current signal analysis needs to be improved with a given method. This is not really justified as from my point of view the method currently under analysis is not really well understood. Not the method per say, but the results it offers.

Response: Many thanks for your comments! In the revised manuscript, we have added some discussion about the results at the Second 5. More features always offer more evidence for diagnosis. The combination of current and vibration features makes the diagnosis more reliable, so different features should be considered. But among the features, as can be seen from the experimental results, compared with the features of vibration signal, the features of the current are not so obvious, which may be due to the selection of the dictionary or the optimization algorithm. Therefore, in the future, dictionary learning algorithms should also be studied, in order to be more suitable for the feature extraction of motor signals. In addition, as you mentioned, more complicated conditions and more PMSM faults need to be considered.

2. In Table 1. Point 5) is A_t missing in the last part of the line before the expression \hat \alpha is inserted from point 4)?

Response: Many thanks for your careful check! We are sorry for our negligence. We modified the formula in Point 5, Table 1 in the revised manuscript.

3. Page 9 (line 194) it is mentioned that normal signals are higher and faulty signals are lower but from Figs. 9 the faulty features appears larger (stronger) than the normal features.

Response: Many thanks for your careful check! We are sorry for our negligence! The fault features are indeed higher than the normal. In the revised manuscript we modified the statement about it.

4. F_i, i={1,..,4} represents the extracted features. Which is what is somewhat vaguely mentioned in the text, but can it be written explicitly at the start of Section 4.3?

Response: Many thanks for your comments! In the revised manuscript, we have added some accurate description about the meaning of F1, F2, F3, F4 at the start of Section 4.3.

Reviewer 2 Report

The manuscript proposes a electrical fault-diagnosis method to detect inter-turn short circuits in electric motors. The approach is based on the feature extraction from current and normal acceleration (vibration) signals. Then, the authors feed a Support Vector Machine (SVM) algorithm with the obtained features to provide a binary fault classification.

The paper is well-written, well-structured and easy to follow. The covered topic is of interest to the target journal.

However, I have a few concerns:

1. I wonder why the paper targets permanent-magnet synchronous motors. Since vibration and current waveform are extracted, probably the same principle can be applied to other motor types. Please clarify.

2. The algorithm relies on vibration and current information. I would like to understand if the algorithm is robust against unwanted disturbances on the measured signals (e.g. harmonic distortion on the current due to supply noise or additional vibrations due to rotor unbalance).

3. In page 9, the following sentences are stated:

"All the first features of normal vibration are higher than 0.4, while the fault features are lower. Besides, the second features of normal vibration are higher than 0.25, while the fault are lower. Similar phenomena can also be found in the current signals, although not so obvious."

However, from Figs. 8 and 9, the opposite is true: fault signal features are larger than those from normal signals. Please correct the text accordingly.

4. It would be interesting to include an FFT of the example vibration and current signals of figures 2 and 3. A comparison of the frequency responses between the normal and the faulty machines could highlight the importance of the proposed method.

5. I noticed very few English errors, I suggest a complete text check to correct those mistakes. 

Author Response

1. I wonder why the paper targets permanent-magnet synchronous motors. Since vibration and current waveform are extracted, probably the same principle can be applied to other motor types. Please clarify.

Response: Many thanks for your comments! In the field of mechanical fault diagnosis for induction motors and gearboxes, the application of sparse representation has been studied. Compared with induction motors, research on permanent magnet synchronous motors is still relatively rare. Since the PMSM replaces the field winding with a permanent magnet, its current and vibration signals are different from the induction motor in the case of fault. Besides, the prior knowledge about it is relatively small. For example, what are the meaning of different harmonic frequencies, what features should be extracted, etc., all of which need to be studied. Therefore, it would be more meaningful to use the sparse representation method for PMSM fault diagnosis .thanks for your suggestion! This method may be changed appropriately to be apply to other motor.

2. The algorithm relies on vibration and current information. I would like to understand if the algorithm is robust against unwanted disturbances on the measured signals (e.g. harmonic distortion on the current due to supply noise or additional vibrations due to rotor unbalance).

Response: Many thanks for your careful check! In the revised manuscript, we have added some explanations about the advantages of sparse representation at the start of Second 2. Sparse representation can extract the basic information from the signal, and avoid the interference of small noise in the signal, such as additional vibration caused by rotor imbalance or distortion of the measurement process.

3. In page 9, the following sentences are stated:

"All the first features of normal vibration are higher than 0.4, while the fault features are lower. Besides, the second features of normal vibration are higher than 0.25, while the fault are lower. Similar phenomena can also be found in the current signals, although not so obvious."

However, from Figs. 8 and 9, the opposite is true: fault signal features are larger than those from normal signals. Please correct the text accordingly.

Response: Many thanks for your careful check! We are sorry for our negligence! The fault features are indeed higher than the normal. In the revised manuscript we modified the statement about it.

4. It would be interesting to include an FFT of the example vibration and current signals of figures 2 and 3. A comparison of the frequency responses between the normal and the faulty machines could highlight the importance of the proposed method.

Response: Many thanks for your comments! We have added some spectrum diagrams obtained by FFT in the revised manuscript that are Figure 3 and Figure 5.

5. I noticed very few English errors, I suggest a complete text check to correct those mistakes.

Response: Many thanks for your comments! We are sorry for our negligence. We removed some grammatical errors and modified some sentences to make the article more fluent.

Round 2

Reviewer 1 Report

The quality of the paper has improved through the later revision. In Section 5 I would avoid using the word “prove” (Experiments prove that the chosen features and the proposed method are feasible.) as the experiment doesn’t deal with variations in operating conditions such as speed, load, noise, separability to other faults etc. thus wording such as “indicate” seems more proper.

There is also a claim that combination of current and vibration is more reliable, but I cannot see that this is supported in the work.

In the discussion it is stated that the features between normal and faulty current signals are not powerful enough for robust diagnosis. Also, in my previous comment I asked for details on what features the algorithm is extracting from the time series data. The reason for this is to compare if the method detects new unique features or extracts features known from previous literature.

Author Response

1. The quality of the paper has improved through the later revision. In Section 5 I would avoid using the word “prove” (Experiments prove that the chosen features and the proposed method are feasible.) as the experiment doesn’t deal with variations in operating conditions such as speed, load, noise, separability to other faults etc. thus wording such as “indicate” seems more proper.

Response: Many thanks for your comments! In the revised manuscript, we have changed the word “prove” into “indicate”.

There is also a claim that combination of current and vibration is more reliable, but I cannot see that this is supported in the work.

Response: Many thanks for your careful check! In principle, more information can make the SVM classification results more accurate. Even if the information is not sharply differentiated, it will not have an error effect on the classification effect basing on the vibration information, but when the vibration features are not so accurate, the current features can provide evidence. However, as you mentioned, it may be not obvious in the result of this work. In the revised manuscript, we have deleted this sentence.

In the discussion it is stated that the features between normal and faulty current signals are not powerful enough for robust diagnosis. Also, in my previous comment I asked for details on what features the algorithm is extracting from the time series data. The reason for this is to compare if the method detects new unique features or extracts features known from previous literature.

Response: Many thanks for your comments! Since the selected dictionary is a DCT dictionary, the features mainly represent the energy performance of the signal in the frequency domain, mainly in the low frequency part, so it can remove the redundant information while retaining the basic information. One of the advantages of sparse representation is that prior knowledge about the specific feature frequency which is important in spectrum analysis is not needed, which facilitates fault diagnosis for the PMSM. The similar idea can be found in mechanical fault diagnosis. We have added some description about the DCT dictionary in Section 2.